# Multidrug-resistant non-typhoidal *Salmonella enterica* from chickens, farmworkers, and environments: One health implications from Northwestern Ethiopia

Azeb Bayu Mengistu[1], Mequanint Addisu Belete[2]*, Habtamu Tassew[3], Haregua Yesigat Kassa[2], Beyenech Gebeyehu Alemu[3], Hailehizeb Cheru Tegegne[1], Demelash Areda[4]

1 Department of Veterinary Science, College of Agriculture and Environmental Science, Debre Tabor University, Debre Tabor, Ethiopia, 2 Department of Veterinary Laboratory Technology, College of Agriculture and Natural Resources, Debre Markos University, Debre Markos, Ethiopia, 3 School of Veterinary Medicine, College of Agriculture and Environmental Sciences, Bahir Dar University, Bahir Dar, Ethiopia, 4 School of Arts and Sciences, Ottawa University, Civic Center Plaza, Surprise, Arizona, United States of America

* mequanentaddisu@gmail.com, mequanint_addisu@dmu.edu.et

## Abstract

Non-typhoidal *Salmonella* are important foodborne zoonotic pathogens closely linked to poultry and poultry products. Despite their public health importance, limited data are available on the epidemiology and antimicrobial resistance patterns of nontyphoidal *Salmonella* in poultry production systems in Ethiopia. This cross-sectional study aimed to estimate the prevalence, identify risk factors, and assess the antimicrobial resistance profiles of *Salmonella enterica* from poultry farms in Bahir Dar city, northwestern Ethiopia. Standard bacteriological methods, PCR-based detection, and serotyping were used to investigate the presence of *Salmonella* in chicken (n = 126), environmental (n = 198), and human (n = 45) samples collected from 22 poultry farms. Antimicrobial susceptibility profiles were determined using the Kirby–Bauer disk diffusion method. Data from questionnaires and Fisher's exact tests were used to identify risk factors associated with the occurrence of *Salmonella*. Nontyphoidal *Salmonella* species were detected on 18.1% (4/22) of the farms. *Salmonella enterica* was recovered from 3.1% (6/198) of environmental boot samples, 3.2% (4/126) of cloacal swabs, and 4.4% (2/45) of human stool samples. Two *Salmonella* serotypes were identified from among 12 *Salmonella* isolates: *S.* Enteritidis (41.6%, 5/12) and *S.* Typhimurium (16.6%, 2/12). All *Salmonella* isolates demonstrated complete resistance to ampicillin (100%) and tetracycline (100%) and exhibited multiple drug resistance patterns, with a high multiple antibiotic resistance index ranging from 0.45 to 0.55. The prevalence of *Salmonella* was significantly associated with the absence of foot baths (p = 0.0096) and the presence of other animal species on the farm (p = 0.026). The demonstrable emergence of multidrug-resistant *Salmonella*

**Data availability statement:** All relevant data are within the paper and its Supporting Information files.

**Funding:** The author(s) received no specific funding for this work.

**Competing interests:** The authors have declared that no competing interests exist.

Enteritidis and *Salmonella* Typhimurium serotypes, alongside key factors driving the prevalence of nontyphoidal salmonellosis on poultry farms in northwestern Ethiopia, underscores the need for improved intervention strategies and ongoing large-scale One Health genomic surveillance to accurately monitor temporal dynamics of *Salmonella* infections and mitigate the rise of multidrug resistance.

## Introduction

*Salmonella* is a significant pathogen responsible for numerous foodborne illnesses and zoonotic diseases [1]. Globally, an estimated 93.8 million cases of nontyphoidal *Salmonella* (NTS) enteric illnesses occur annually, resulting in approximately 155,000 human deaths. In contrast, Africa reported a lower estimated burden, with 2.4 million cases and 4100 deaths from NTS gastroenteritis across the World Health Organization (WHO) subregions [2]. However, in Sub-Saharan Africa, NTS frequently causes invasive diseases (iNTS) characterized by severe bacteremia and extraintestinal illness, particularly affecting children and immunocompromised adults [3]. Nontyphoidal *Salmonella* is also a leading cause of invasive infections and diarrhea among children and adults in Ethiopia, with a pooled prevalence rate of 57.9% [4].

The complex epidemiology of *Salmonella* and the challenges associated with its control result in substantial economic losses, hindering the sustainable growth of the poultry industry. Among the over 2500 *Salmonella* serotypes, *Salmonella* Enteritidis (*S. Enteritidis*) and *Salmonella* Typhimurium (*S. Typhimurium*) are particularly problematic in poultry and poultry products [5]. Since 1980, documented pandemics and outbreaks have consistently underscored the crucial role of *S.* Enteritidis and *S.* Typhimurium in the overall burden of *Salmonella* infections in poultry farming systems and public health settings across North America, South America, Europe, Asia, and Africa [6,7]. These two serotypes exhibit remarkable adaptability to adverse environments and can persist in diverse ecological niches outside poultry farms, including food production environments and food products [8].

Food-producing animals remain the principal source of human *Salmonella* infections worldwide [9]. For instance, surveillance data from the United States in 2017 revealed that 29% of *Salmonella* outbreaks and 34% of associated illnesses primarily originated from contaminated poultry, making it the leading source among terrestrial animals. *S.* Enteritidis was the most prevalent serotype, reported in 27 outbreaks, followed by *S.* Typhimurium serotype, reported in 14 outbreaks [10]. However, recent studies indicate that direct contact with infected birds or indirect exposure to contaminated environments can also contribute to human *Salmonella* infections, emphasizing the critical importance of the One Health framework [11,12]. In North America and West Africa, genomic surveillance of *Salmonella* species has identified strains closely related to humans, poultry, and the environment, sharing similarities in phylogenetics, core genome, pangenome, and virulence markers [13,14]. In Korea, a clonal comparison of *S.* Enteritidis and *S.* Typhimurium isolates from humans and broiler chickens revealed identical genetic types, demonstrating the potential for zoonotic

transmission from poultry to humans [15]. Moreover, recent data in Ethiopia indicate an increasing overlap between *Salmonella* strains isolated from humans and poultry fecal droppings, highlighting the need to explore the significance of *Salmonella* isolation across various sample sources [16].

The emergence and dissemination of antimicrobial resistance (AMR) in bacteria present significant global public health challenges, impacting human, animal, and environmental health [17,18]. In low-income settings, factors such as the misuse or overuse of antibiotics, limited awareness of AMR, and close contact between humans and animals contribute to the proliferation of drug-resistant bacteria. Poultry, in particular, acts as a reservoir for the dissemination of antibiotic-resistant NTS strains through the food chain [19,20]. A comparative genomic analysis conducted in Mexico revealed that poultry harbors multidrug-resistant nontyphoidal *Salmonella* (MDR-NTS) with AMR genotypes highly similar to those observed in human clinical isolates [21]. Similarly, in Senegal, combined antimicrobial susceptibility testing and whole genome analysis of *Salmonella* isolates from humans and chickens indicated the potential transmission of MDR serotypes from chickens to humans, identifying broilers as a source of antimicrobial resistance [22].

Numerous studies conducted in Ethiopia have documented a significant increase in the prevalence of MDR-NTS within the poultry sector. Eguale [23] reported a high prevalence of multidrug-resistant (MDR) *S.* Saintpaul and *S.* Kentucky serotypes on poultry farms in central Ethiopia. Similarly, other studies have disclosed that 69.5% to 96.77% of *Salmonella* isolates from poultry farms in urban and peri-urban areas of central Ethiopia exhibited MDR [16,24,25]. Notably, a high proportion, 93.4% (42/45) of *Salmonella* isolates from the poultry industry in southern Ethiopia were found to be MDR [26]. However, the incidence and impact of these MDR pathogens remain poorly understood, primarily due to the absence of robust, coordinated laboratory and epidemiological surveillance systems [27].

Previous studies investigating the occurrence, serotype distribution, and antimicrobial susceptibility profile of NTS on poultry farms in Ethiopia have been limited in both number and geographical scope. Although the region is a major poultry producer, the characteristics and antibiotic resistance profile of NTS on poultry farms in Bahir Dar, located in northwestern Ethiopia, remains largely unexplored. Furthermore, research on non-typhoidal salmonellosis from a One Health perspective—which emphasizes the interconnectedness of animal, human, and environmental health is scarce in this region. Therefore, this study aimed to investigate the prevalence, identify potential determinant factors, and determine the most common serotypes and antimicrobial resistance patterns of NTS isolated from poultry, humans, and farm environments in Bahir Dar city, northwestern Ethiopia.

## Materials and methods

### Ethical considerations

The study protocol (Ref. No: BU/ECRC/1/112/1.3/2020) was reviewed and approved by the institutional Ethical and Environmental Considerations Review Committee of Bahir Dar University. Written informed consent was obtained from study participants after a thorough explanation of the purpose of the study.

### Description of the study area

The study was conducted in Bahir Dar, the capital administrative city of Amhara National Regional State, in the northwestern part of Ethiopia. Geographically, Bahir Dar is situated between 11.29° to 11.38° north latitude and 37.23° to 37.36° east longitude, with an average elevation estimated to be 1810 m above sea level. The city experiences an average annual temperature of 20.85 °C and precipitation of 1419 mm [28]. In recent years, commercial poultry farming has seen significant growth in urban and peri-urban areas, providing a source of income through the sale of eggs and live birds. However, the proximity of many of these farms to residential areas raises concerns about the potential transmission of pathogens and the spread of antibiotic resistance.

## Study design, sampling, and study samples

A cross-sectional study was conducted over nine months from November 2020 to July 2021. The study population consisted of chickens of different age groups (<2 months, 2–6 months, 7–12 months, > 12 months) and human contacts who owned or worked on selected poultry farms. Twenty-two poultry farms were randomly selected from the list provided by the Bahir Dar Agricultural Development Office. These included 10 small farms (≤1000 birds) and 12 large farms (>1000 birds). The number of live birds sampled per farm was proportionally allocated. Farms without official records and lacking contact with livestock extension services or veterinary officers were excluded from the study. Data collection involved direct personal observations and semi-structured questionnaire surveys (S2 File) to gather information on farm size, bird type, bird age, history of antimicrobial use, and other poultry farm management practices. The questionnaire was pretested and administered through face-to-face interviews with one designated farm attendant from each of the 22 farms. For the current study, different samples were collected, including animal samples (chicken cloacal swabs), pooled environmental samples (boot swabs, poultry feed, and poultry drinking water), and pooled human samples (stool samples from farm handlers) (Fig 1).

## Sample collection

A total of 369 samples were collected, comprising 126 cloacal swab specimens from chickens, 198 poultry-associated environmental samples, and 45 stool samples from volunteer farm attendants. The environmental samples included water (n = 66), feed (n = 66), and boot swabs (n = 66), with three pooled samples collected per farm. Cloacal swab samples were collected using sterile cotton swabs containing 10 mL of buffered peptone water (BPW) (Oxoid, Basingstoke, UK) by gently rotating the swab within the cloaca. Each bird was sampled only once. Boot swab samples were collected by wearing sterile plastic boots and walking on the poultry house floor. Poultry feed (5 g) and drinking water (5 mL) samples were collected from each farm using sterile zippered plastic bags and test tubes, respectively [29]. Approximately 1 g of stool sample was collected from human volunteers using sterile stool cups with applicators [30]. All samples were transported within two hours under cold chain conditions in an icebox to the microbiology laboratory unit of the Amhara Public Health Institute (APHI). Samples were either processed immediately or stored at 4 °C for a duration of 4–48 hours.

**Isolation and molecular identification of *Salmonella* species and serotypes.** The isolation and identification of non-typhoidal *Salmonella* were conducted according to standard microbiological guidelines [31]. Briefly, cloacal and boot swabs, feed (5 g), water (5 mL), and a stool (1 g) sample were pre-enriched in 10 mL, 45 mL, and 9 mL of BPW (Oxoid, Basingstoke, UK), respectively. All samples were vortex-mixed and incubated at 37 °C for 16–20 hours. Subsequently, 0.1 mL and 10 mL aliquots of the pre-enriched cultures were selectively enriched by inoculating them into 9.9 mL of Rappaport–Vassiliadis (RV) broth (Oxoid, Basingstoke, UK) at 41.5 °C for 24 hours and 10 mL of Muller–Kauffmann tetrathionate (MKTT) broth (HiMedia, Maharashtra, India) at 37 °C for 24 hours. Enriched cultures from both RV and MKTT broths were streaked onto xylose-lysine-desoxycholate (XLD) agar plates (HiMedia, Maharashtra, India) and incubated at 37 °C for 24–48 hours. Following incubation, the plates were examined for microbial growth, and colonies suspected to be *Salmonella* (characterized by colorless to red/pink with and without black centers) were selected. These colonies were subcultured onto nutrient agar (NA) plates (HiMedia, Maharashtra, India) and incubated at 37 °C for 18–24 hours. Isolates were further identified using a series of biochemical tests, including the indole test, triple sugar iron (TSI) test, citrate utilization test, urease test, lysine iron agar test, and sulfide-indole-motility (SIM) test, as previously described [23] and interpreted in accordance with the International Organization for Standardization (ISO) guidelines [31].

Genus-level identification of presumptive *Salmonella* isolates was performed using uniplex PCR targeting the histidine transport operon gene, as previously described [32]. Isolates confirmed as *Salmonella* spp. were further subjected to serotype identification by detecting the *S. Typhimurium*-specific gene (*spy*) [33] and *S. Enteritidis*-specific gene (*sdf1*) [34] (Fig 1).

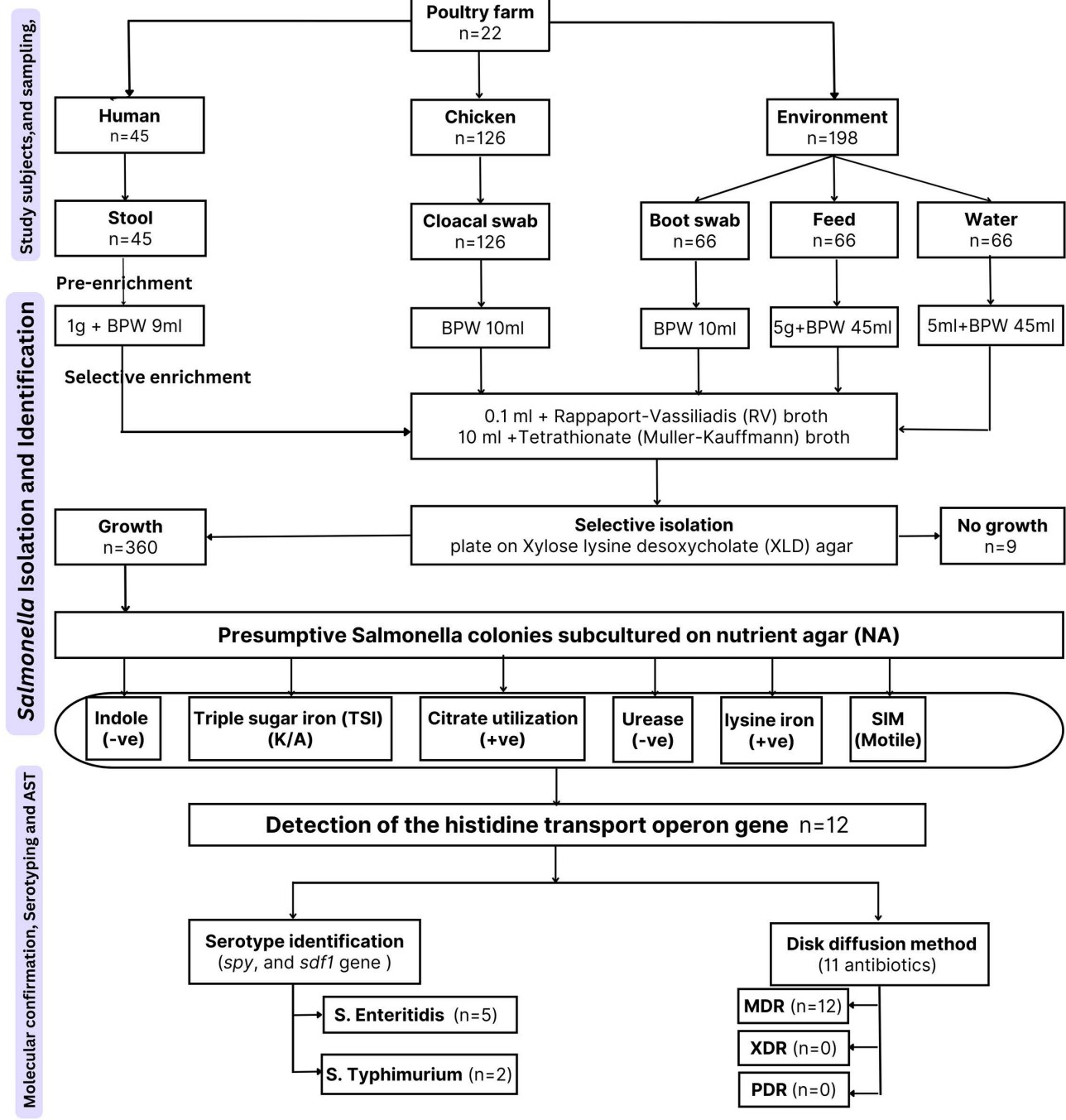

**Fig 1. Schematic flowchart illustrating the study design, including sample collection, *Salmonella* isolation, biochemical and molecular confirmation, serotype identification, and antimicrobial susceptibility testing.**

In brief, bacterial DNA was extracted from overnight cultures grown on nutrient agar (NA) using a boiling lysis method. The quality and concentration of the extracted DNA were assessed using NanoDrop 2000/2000c spectrophotometers (Thermo Scientific, Boston, MA, USA). PCR assays were performed using a GeneAmp PCR System 2400 thermocycler

(PerkinElmer, Shelton, CT, USA) in a 25-µL reaction volume. Each reaction contained 1 µL of DNA template, 12.5 µL of 2X PCR Master Mix (Promega, Madison, WI, USA) (containing 50 U/mL of Taq DNA polymerase, 3 mM MgCl2, and 400 µM each of dNTP mix), 0.5 µL of 0.2 µM of each primer (Bioneer, Daejeon, South Korea), and 10.5 µL of nuclease-free water.

The PCR cycling conditions included an initial denaturation at 94 °C for 5 minutes, followed by 30 cycles of denaturation (94 °C for 30 seconds), annealing (as shown in Table 1) for 30 seconds), and extension (72 °C for 30 seconds), with a final extension step (72 °C for 7 minutes). PCR products were analyzed by electrophoresis on 1.5% agarose gels (Bio Basic, Markham, ON, Canada) in 1 × TAE buffer, stained with 1 µg/mL ethidium bromide. Gels were visualized and photographed using a gel documentation system (Bio-Rad, Hercules, CA, USA). A 100 bp DNA ladder (HiMedia, Maharashtra, India) was used as a molecular size marker.

## Antimicrobial susceptibility testing

Antimicrobial susceptibility testing was performed using the Kirby–Bauer disk diffusion method on Muller–Hinton Agar (MHA) (HiMedia, Maharashtra, India) according to the Clinical and Laboratory Standards Institute (CLSI) guidelines [35]. All *Salmonella* isolates were evaluated for susceptibility to 11 antibiotics representing eight antimicrobial classes: penicillins (ampicillin (10 µg)); cephalosporins (cephalothin (30 µg)); phenicols (chloramphenicol (30 µg)); fluoroquinolones (ciprofloxacin (5 µg), nalidixic acid (30 µg)); aminoglycosides (gentamycin (10 µg), streptomycin (10 µg), kanamycin (30 µg)); tetracyclines (tetracycline (30 µg)); sulfonamides (sulfisoxazole (1000 µg) and the folate pathway antagonist (sulfamethoxazole-trimethoprim (23.75/1.25 µg) (Oxoid, Basingstoke, UK). Antibiotic susceptibility was determined based on the breakpoints defined in CLSI M100 [35] and CLSI VET08 [36]. Resistant *Salmonella* isolates were categorized as multidrug-resistant (MDR), extensively drug-resistant (XDR), and pan-drug-resistant (PDR) according to the criteria outlined by Magiorakos et al [37]. The multiple antibiotic resistance index (MAR) was calculated using the following formula: $MAR = a/b$, where a represents the number of antibiotics to which the isolate was resistant, and b represents the total number of antibiotics tested in this study, as described in [38] (Fig 1).

## Quality control and quality assurance

To ensure data quality and the reliability of the study findings, sample collection, transport, and storage strictly adhered to standard operating procedures. The sterility of the prepared media was verified by incubating plates for 24–48 h at 37 °C. To assess the accuracy of screening, confirmatory, and disk diffusion antimicrobial susceptibility tests, *S.* Enteritidis ATCC 13076 and *S.* Typhimurium ATCC 13311 (positive control) and *Escherichia coli* ATCC 25922 (negative control) were included alongside the tested isolates. These control strains were obtained from the Bacteriology, Parasitology, and Zoonosis Research Unit at the Ethiopian Public Health Institute (EPHI) in Addis Ababa, Ethiopia.

**Table 1. Oligonucleotide primer sequences and their annealing temperature used in the polymerase chain reaction.**

| Primer use (Target Gene) | Nucleotide sequence (5′ → 3′) | Amplicon Size (bp) | Annealing (°C) | Reference |
|---|---|---|---|---|
| *Salmonella enterica* | F: ACTGGCGTTATCCCTTTCTCTGGTG | 496 | 54 | [32] |
| (*histidine transport operon*) | R: ATGTTGTCCTGCCCCTGGTAAGAGA | | | |
| *S.* Typhimurium | F: TTATTCACTTTTTACCCCTGAA | 401 | 52 | [33] |
| (*spy*) | R: CCCTGACAG CCGTTAGATATT | | | |
| *S.* Enteritidis | F: TGTGTTTTATCTGATGCAAGAGG | 304 | 55 | [34] |
| (*sdf1*) | R: TGAACTACGTTCGTTCTTCTGG | | | |

## Data analysis

Data were initially entered and managed in Microsoft®Excel® 2019 (version 16.0.10414.20002; Microsoft Corporation, Redmond, WA, USA). Prior to statistical analyses, the dataset underwent preprocessing, which included accuracy verification, removal of duplicate entries, handling missing values, and consistency checks across recorded variables to ensure quality. The cleaned dataset was then exported to STATA version 13.0 (StataCorp, College Station, TX, USA) for statistical analysis. Descriptive statistics were used to summarize farm and sample-level characteristics. The prevalence of *Salmonella* on poultry farms was estimated as a proportion of farms with at least one positive sample, either from poultry or poultry-associated sources, relative to the total number of farms sampled. Associations between the isolation frequency of *Salmonella* and putative risk factors were assessed using Fisher's exact test, due to small sample sizes and the presence of expected cell frequencies less than five. Exact 95% confidence intervals were calculated using the binomial distribution. A p-value of <0.05 was considered statistically significant.

## Results

### Farm characteristics

Poultry farms in this city are organized under Small and Micro-Enterprise offices. The majority (68.2%, 15/22) of farms focus primarily on egg production. All farms employed floor housing systems and implemented the all-in-all-out management principle, ensuring thorough cleaning and disinfection of poultry houses between flocks. Common clinical signs observed in these flocks included greenish-watery diarrhea, drooping heads, loss of appetite, bloody diarrhea, wing droop, and respiratory symptoms such as sneezing, coughing, and nasal discharge. Many farms also raised other domestic animals, including cattle and sheep, and lacked dedicated rodent control programs. Significant discrepancies were observed among the farms in feeding practices, water sources, and overall management strategies (Fig 2). While 32% of farms relied solely on commercially formulated feeds, 31.9% utilized a combination of commercial and on-farm-produced feeds. A concerning issue was the occurrence of sudden and unexpected flock mortality without any preceding signs of illness. Furthermore, a significant number of farm workers did not use adequate personal protective equipment (PPE).

### Prevalence of *Salmonella* isolates in poultry farms

Nontyphoidal *Salmonella* was identified in 18.2% (4/22) of poultry farms, including 20% (3/15) of layer farms and 14.3% (1/7) of broiler farms (Table 2). The prevalence of *Salmonella* (33.3%, 4/12) was higher on large-scale farms compared to small-scale farms. Regarding the age of the flock, a higher proportion of positive samples for *Salmonella* spp. was observed in chickens aged 2–6 months (23.1%, 3/13) (Table 2). The study demonstrated a statistically significant (p = 0.0096) association between the use of disinfected boots and a lower prevalence of *Salmonella* on the farms. Intriguingly, the presence of other animal species within the poultry farms was identified as a significant risk factor for *Salmonella* positivity (p = 0.026). Production type, flock size, and flock age were not significantly associated with *Salmonella* prevalence on poultry farms. The characteristics of the farms and the extent of *Salmonella* prevalence in relation to contributing risk factors are summarized in Table 2.

**Occurrence and serotypes of nontyphoidal *Salmonella* isolates based on sources.** Of the samples tested, 3.3% (12/369) were positive for *Salmonella* species (S1 Fig). The prevalence of *Salmonella* in environmental boot samples (9.1%, 6/66) was significantly higher than in chicken cloacal samples (3.2%, 4/126) and human stool samples (4.4%, 2/45) (p = 0.031) (Table 3). PCR assays using serotype-specific genetic markers revealed the presence of two serotypes. Regardless of the isolate source, the most prevalent serotype was *S*. Enteritidis at 41.6% (5/12), followed by *S*. Typhimurium at 16.6% (2/12). Untyped *Salmonella* strains accounted for 41.6% (5/12). *S. Enteritidis* was detected in all sample sources, with the highest prevalence in chicken samples. *S. Typhimurium* was only isolated from environmental samples (Table 3). All poultry feed and drinking water samples tested negative for *Salmonella* spp., *S*. Enteritidis, and *S*. Typhimurium. The overall and sample-wise prevalence of *Salmonella* is summarized in Table 3.

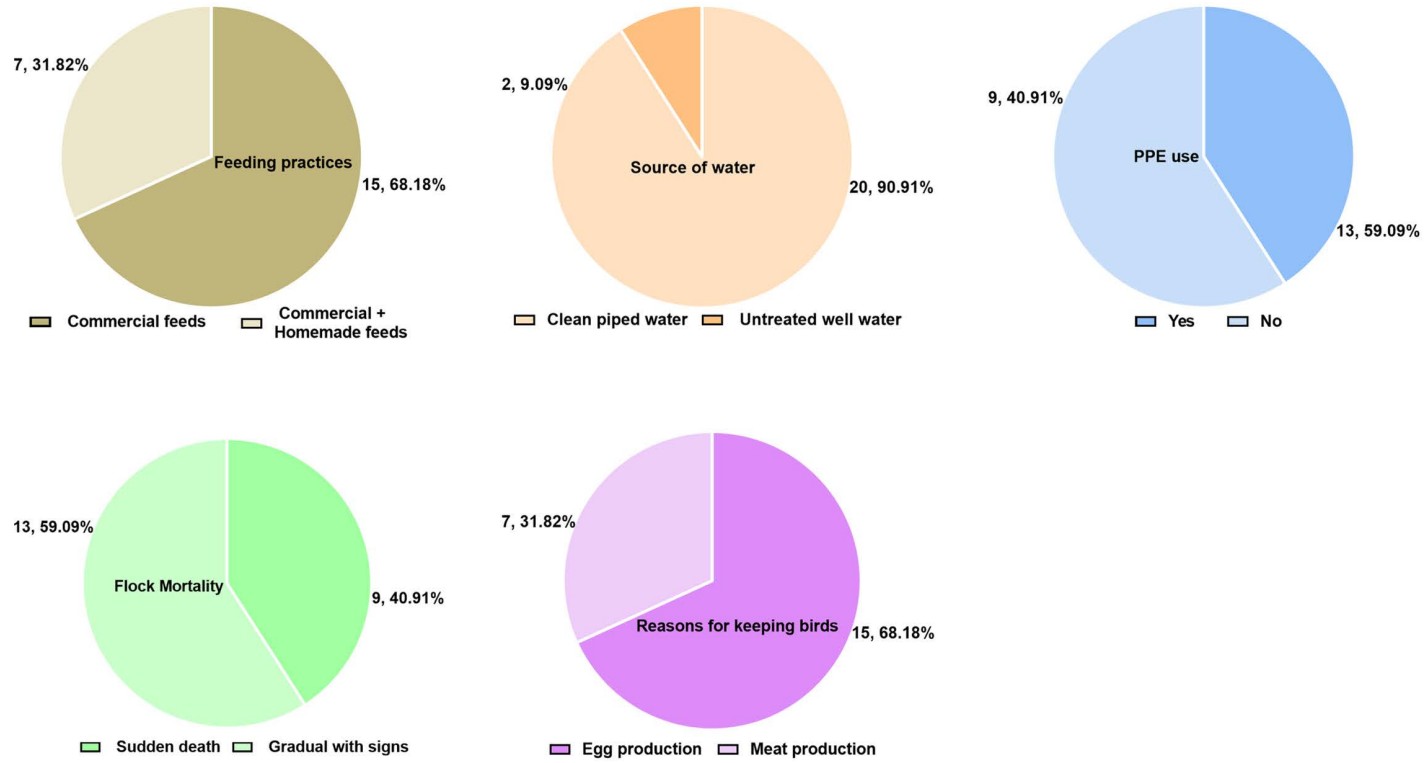

**Fig 2. Feeding practices, water sources, biosecurity measures, and mortality rates on 22 poultry farms in Bahir Dar, northwestern Ethiopia.**

## Antimicrobial susceptibility profile of *Salmonella* isolates

Fig 3 summarizes the results of the antimicrobial susceptibility testing of 11 antimicrobial agents belonging to eight different classes of antibiotics. *Salmonella* isolates from humans, chickens, and the environment exhibited 100% resistance to tetracycline and ampicillin, followed by 80% and 60% resistance to streptomycin and cephalothin, respectively. Moderate resistance to sulfisoxazole, nalidixic acid, and kanamycin was observed at a rate of 40%. In contrast, all isolates showed 100% sensitivity to gentamicin and ciprofloxacin (Fig 3a).

The resistance rate for the antibiotic agents against isolates from humans, chickens, and the environment was as follows: AMP (100%, 100%, and 100%), CEF (100%, 0%, and 100%), CHL (0%, 50%, and 0%), NAL (0%, 100%, and 0%), STR (100%, 50%, and 100%), KAN (0%, 50%, and 50%), TET (100%, 100%, and 100%), SFX (100%, 0%, and 50%), and SXT (0%, 50%) and 50%). Results revealed that *Salmonella* isolates from chickens exhibited unique resistance patterns, including resistance to nalidixic acid (n = 4) and chloramphenicol (n = 2). In addition, resistance to ampicillin, streptomycin, and tetracycline was observed across all sample sources (Fig 3b).

All *Salmonella* isolates (100%) were classified as MDR, exhibiting resistance to three or more classes of antibiotics. *Salmonella* isolates from humans and the environment exhibited a common antimicrobial resistance pattern (33.3%), characterized by resistance to AMP, CEF, STR, TET, and SXT. The MAR index for all isolates exceeded 0.2, indicating a high level of resistance. The data showed that *S.* Typhimurium and *S.* Enteritidis isolates exhibited the highest MAR index value of 0.55, with six similar phenotypic antibiotic resistance patterns (AMP, CEF, STR, KAN, TET, and SFX). The most frequent MAR index observed was 0.45 (resistant to five out of eleven selected antibiotic agents) (Fig 3c). Extensively drug-resistant or pan-drug-resistant isolates were not recorded in the present study.

## Discussion

The poultry and livestock sectors are major contributors to the emergence and spread of antibiotic resistance, primarily due to the widespread and indiscriminate use of antibiotics for disease prevention and growth promotion. Poultry, in particular, serves as a significant reservoir of resistant bacteria and genes, posing a substantial risk to human health [39,40]. In resource-limited countries, including Ethiopia, antimicrobial surveillance systems that evaluate the level of antibiotic

**Table 2. Variation in *Salmonella* prevalence based on selected risk factors among poultry farms in Bahir Dar, Ethiopia.**

| Factors | No. of farms | No. of positive | Prevalence in % (95% CI) | p-value |
|---|---|---|---|---|
| Commodity type | | | | 1.0000 |
| Layers | 15 | 3 | 20.0 (4.3–48.0) | |
| Broilers | 7 | 1 | 14.3 (0.37–57.7) | |
| Flock size | | | | 0.0964 |
| Small a | 10 | 0 | 0.0 (0.0–30.8) | |
| Large b | 12 | 4 | 33.3 (9.9–65.1) | |
| Age of chickens | | | | 0.9572 |
| <2 months | 2 | 0 | 0.0 (0.0–77.6) | |
| 2–6 months | 13 | 3 | 23.1 (5–53.8) | |
| 7–12 months | 5 | 1 | 20.0 (0.5–62.4) | |
| >12 months | 2 | 0 | 0.0 (0.0–77.6) | |
| Antimicrobial use | | | | 0.5352 |
| Yes | 17 | 4 | 23.5 (6.8–49.9) | |
| No | 5 | 0 | 0.0 (0.0–52.2) | |
| Foot bathing practice | | | | 0.0096* |
| Yes | 14 | 0 | 0.0 (0.0–21.5) | |
| No | 8 | 4 | 50.0 (15.7–84.3) | |
| Presence of rodents and insects | | | | 0.5352 |
| Yes | 17 | 4 | 23.5 (6.8–49) | |
| No | 5 | 0 | 0.0 (0.0–52.2) | |
| Presence of other livestock | | | | 0.0260* |
| Yes | 2 | 2 | 100 (34.2–100) | |
| No | 20 | 2 | 10 (1.2–32.0) | |
| Supplementation of feed additives | | | | 1.0000 |
| Yes | 17 | 3 | 17.6 (3.8–43.4) | |
| No | 5 | 1 | 20 (0.5–71.6) | |

Small a ≤1000; Large b >1000 birds, * = p ≤ 0.05.

**Table 3. Distribution of the most common *Salmonella* serotypes based on sample sources from poultry farms in Bahir Dar, Ethiopia.**

| Host | Samples | No. of Samples | *Salmonella* spp. | *S.* Enteritidis | *S.* Typhimurium | Untyped *Salmonella* |
|---|---|---|---|---|---|---|
| Chicken | Cloacal | 126 | 4(3.2%) | 2 (1.9%) | – | 2(1.6%) |
| Enviroment | Boot | 66 | 6 (9.1%) | 2(3.0%) | 2 (3.0%) | 2(3.0%) |
| | Feed | 66 | – | – | – | – |
| | Water | 66 | – | – | – | – |
| Human | Stool | 45 | 2(4.4%) | 1 (2.2%) | – | 1(2.2%) |
| | Total | 369 | 12(3.3%) | 5(1.4%) | 2 (0.54%) | 5 (1.4%) |

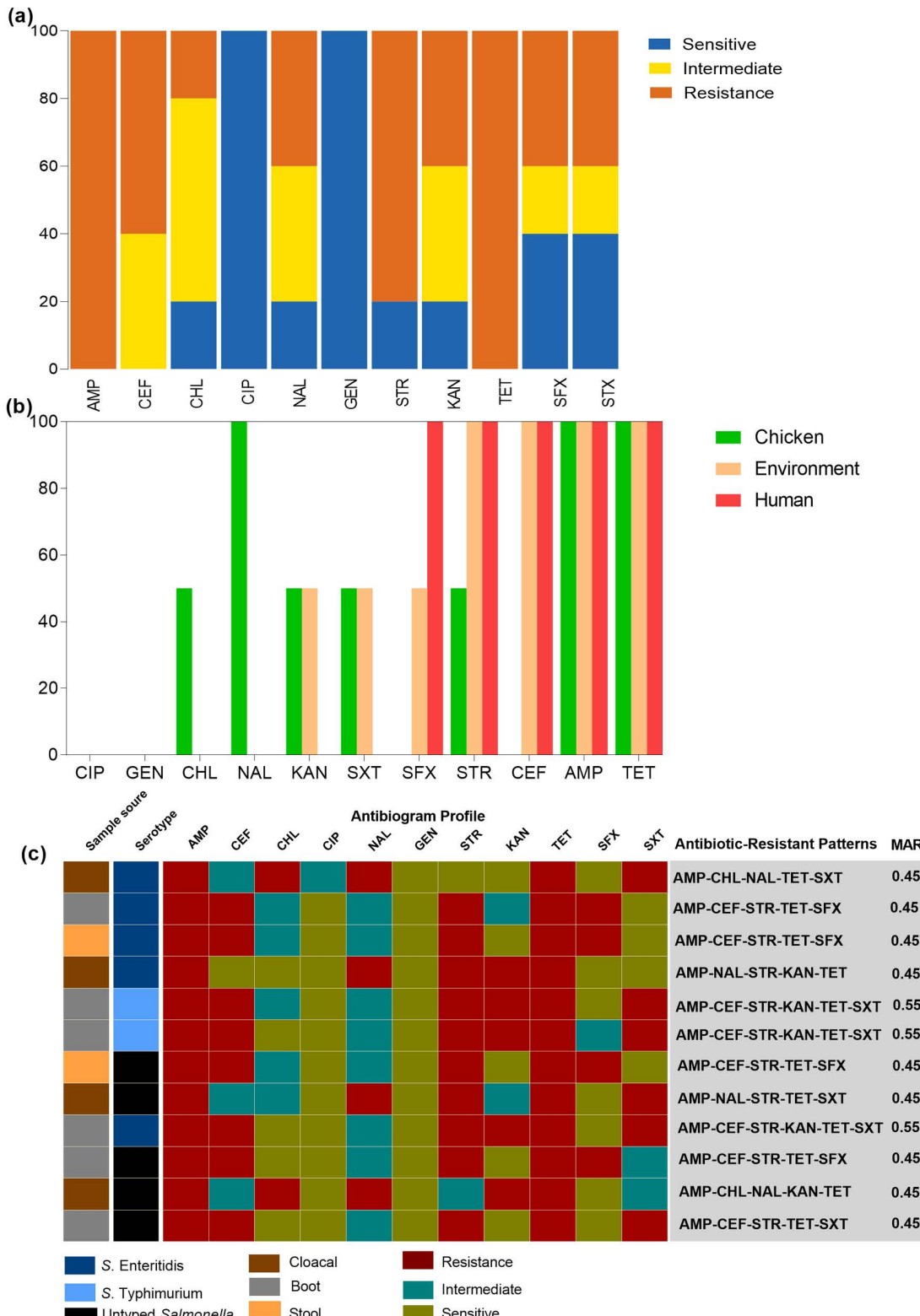

**Fig 3. The phenotypic antimicrobial susceptibility profiles of *Salmonella* isolates from humans, chickens, and the environment in Bahir Dar, Ethiopia. (a)** A column chart illustrates the antibiotic resistance rates observed among *Salmonella* isolates. **b)** The disparities in antibiotic resistance

rates among *Salmonella* isolates from humans, chickens, and the environment. **(c)** Phenotypic antibiotic resistance patterns and multiple antibiotic resistance index of *Salmonella* isolates. AMP = ampicillin, CEF = cephalothin, CHL = chloramphenicol, CIP = ciprofloxacin, NAL = nalidixic acid, GEN = gentamycin, STR = streptomycin, KAN = kanamycin, TET = tetracycline, SFX = sulfisoxazole, SXT = trimethoprim/sulfamethoxazole.

resistance among priority pathogens are largely focused on healthcare facilities and human specimens [41]. This narrow scope hinders a comprehensive understanding of antibiotic resistance and limits the effectiveness of control strategies.

In this study, we employed both conventional and molecular methods to investigate the prevalence and antibiotic resistance patterns of NTS isolates from poultry farms. To the best of our knowledge, this is the first study to document the incidence of NTS on poultry farms in northwestern Ethiopia. Our findings highlight that the farm environment, chickens, and farm workers in contact with poultry are potential sources of *Salmonella* infection. The study identified two circulating *Salmonella* serotypes on the investigated poultry farms, with the isolated NTS strains demonstrating multidrug resistance.

This cross-sectional study conducted on commercial farms revealed a direct effect of flock size, the use of boot dips, and the presence of other livestock on the occurrence of nontyphoidal Salmonellosis. The study provides a foundation for better understanding the transmission dynamics of infections and antibiotic resistance by examining three distinct sample types. These findings are crucial for effectively addressing AMR issues and guiding targeted prevention and control measures against *Salmonella* infection at national, regional, and global scales.

A recent study conducted in Bangladesh [42] and Uganda [43] reported a similar prevalence of NTS on poultry farms. In contrast, the current findings indicate a lower prevalence compared to comparable studies conducted in Nigeria [44], Algeria [45], and Nepal [46]. *Salmonella* contamination was observed in 34.4%, 43.6%, and 55% of sampled farms, respectively. Another cross-sectional retrospective study in Brazil found a 32.1% prevalence of *Salmonella* across various commercial poultry sheds [47]. However, the prevalence reported in this study is slightly higher than the previous findings, including a 14.6% prevalence in poultry farms in central Ethiopia [23], 10. 6% in Indian livestock samples (including poultry cloacal swabs) [48], and 8% in French broiler-chicken flocks [49]. This study observed a higher prevalence of *Salmonella* spp. in the layer farms compared to the broiler farms. This finding aligns with another study that reported a higher prevalence of *Salmonella* in layers (46.2%) compared to broilers (41.3%) [50]. The higher prevalence of *Salmonella* infection observed in laying hens may be attributed to the physiological stress associated with layers during egg production and molting. Such stress significantly impairs the immune response of layers, thereby increasing their susceptibility to *Salmonella* infection [51,52].

In this study, chickens aged 2–6 months showed a higher proportion of *Salmonella* spp. positive samples compared to older chickens. Chickens under two weeks of age are particularly susceptible to gastroenteritis and systemic disease caused by *Salmonella*. In contrast, adult hens often become asymptomatic carriers of *Salmonella enterica*. Although they show no overt symptoms, adult hens intermittently shed the bacteria in their feces [53]. This study found no significant association between the prevalence of *Salmonella* on poultry farms and factors such as production type and age. These findings are consistent with previous studies conducted in Ethiopia, which similarly reported no statistically significant differences in *Salmonella* isolation rates across different production systems and age groups of birds [50,54].

This study demonstrates a higher prevalence of *Salmonella* on large-scale poultry farms. The findings suggest that farms rearing larger flocks are at a higher risk of *Salmonella* contamination compared to those rearing smaller flocks. Indeed, evidence implies that flock size is a critical risk factor for the escalation of *Salmonella* contamination and infection [55,56]. Poultry farms with high flock densities often encounter challenges related to overcrowding. Overcrowding can increase stress levels, reduce feed intake, and impair the immune response of birds. Massive flocks of chickens can also complicate the implementation of strict biosecurity measures and effective farm management practices. These factors collectively contribute to the increased occurrence and spread of *Salmonella* infection within flocks [57,58]. Studies from Korea [59], Bangladesh [60], and Nigeria [61], also report a higher incidence of *Salmonella* spp. on poultry farms with large flocks.

The study identified a significant correlation between the use of disinfected boots, the presence of other livestock, and the prevalence of *Salmonella* on poultry farms. The external environment surrounding poultry houses emerged as a key pathway for the introduction and persistence of pathogens. These areas, often accessed by farm workers and visitors via their boots, can act as conduits for pathogen transmission. The implementation of boot dips containing disinfectant is crucial for mitigating the risk of *Salmonella* contamination and maintaining a safe poultry farming environment [62–64].

Furthermore, the findings suggest that other animals may serve as a source of nontyphoidal salmonellosis on poultry farms and their surrounding environments. *Salmonella* bacteria commonly reside in the gastrointestinal tracts of various animals, including livestock such as cattle, sheep, and pigs. These bacteria can be shed in the feces of both infected and healthy animals, making them potential reservoirs and sources of cross-contamination. Transmission can occur through direct contact with animals or indirectly via contaminated feed, water, equipment, or the surrounding environment [65,66]. These results align with studies conducted in Uganda [43], Algeria [45], and Nigeria [61], which similarly reported a significant increase in the risk of *Salmonella* infection on farms where other animal species were present. Consequently, further research is necessary to comprehensively explore the role of other animal species in the transmission of *Salmonella* on poultry farms. Such investigations could involve the identification of serotypes associated with these animals.

This study observed a low prevalence of *Salmonella* spp. in poultry-related samples compared to findings from other regions of Ethiopia. For instance, Abdi et al. [26] reported a 16.7% recovery rate of *Salmonella* from poultry and environmental samples in southern Ethiopia, while Asfaw Ali et al [67] reported a 14.6% prevalence in the cecal contents of exotic chickens in Debre Zeit. A study conducted on the East Coast of Peninsular Malaysia identified varying prevalence rates across different sources. Fecal samples exhibited the highest positivity rate at 59.5%, followed by cloacal swabs at 46.3%. Notably, sewage and tap water samples demonstrated significantly lower positivity rates [68]. The *Salmonella* prevalence of 2.8% in Modjo and Adama (16), and 2.65% in Jimma, Ethiopia [69] are comparable to the findings of this study. In northern Poland, the prevalence of *Salmonella* spp in broiler chickens has gradually decreased from 2.19% in 2014 to 1.1% in 2015 and 1.22% in 2016 [70].

Interestingly, environmental boot samples demonstrated a higher prevalence of *Salmonella* than chicken cloacal swabs and human stool samples. This finding highlights the critical role of inadequate hygienic and poor biosecurity practices on poultry farms in the study area and suggests that boot or sock swabs are among the most effective methods for detecting *Salmonella* spp. in these environments [71]. This aligns with a previous study conducted in Argentina, which also found that *Salmonella* spp. was most frequently detected on boot swabs, followed by fecal samples [72]. Additionally, the presence of pests such as cockroaches, rodents, and insects may significantly contribute to the dissemination of *Salmonella* within poultry farms. These pests can serve as carriers of pathogens, leading to high levels of contamination in both poultry and the nearby environment [73]. A concurrent study by Adesiyun et al. [57] revealed that 90% of farms with rat infestations exhibited the highest *Salmonella* contamination rates in their environment.

This study observed a lower recovery rate of *Salmonella* spp. from cloacal swabs, likely due to fluctuating bacterial concentrations in the cloacal openings, which are influenced by factors like age, stress, and health status. This inconsistency makes it challenging to isolate *Salmonella* from cloacal samples. Moreover, cloacal samples may contain traces of feces or intestinal content, further complicating detection. Therefore, testing a large number of birds is essential to account for the intermittent shedding of *Salmonella* in feces and increase the likelihood of detecting its presence on farms [72,74].

Regarding stool samples from contact workers, this study reported a higher detection rate (6.7%) compared to a previous study in Adama, Ethiopia (2.8%) [16]. In contrast, feed and drinking water samples tested negative for *Salmonella,* consistent with the findings of Djeffal et al. [45] and Abunna et al. [54]. The observed variation in the *Salmonella* prevalence can be attributed to several factors, including geographical location, sample sizes, sample origin, sampling methods, and laboratory techniques. Additionally, the age of chickens, flock breed, farming systems, and biosecurity measures significantly influence the occurrence of *Salmonella* on poultry farms [75].

Poultry can be infected with various *Salmonella* serotypes, with *S*. Enteritidis and *S*. Typhimurium posing the highest public health concern. In recent years, the prevalence of *S*. Typhimurium has remained relatively stable across different farmed species, while *S*. Enteritidis has shown a significant increase in infections among both poultry and humans [76]. Recent reports of infections and foodborne outbreaks are predominantly associated with contamination from multiple sources of poultry and poultry products contaminated with *S*. Enteritidis [77].

This study also identified *S*. Enteritidis and *S*. Typhimurium as the predominant serotypes. These serotypes have similarly been reported as the most prevalent serotypes in humans, poultry, and animal-derived food samples from Egypt [78], Pakistan [79], China [80], and Malaysia [81]. However, contrary to these findings, a study from Nigeria indicated a limited role for *S*. Enteritidis and *S*. Typhimurium, with their absence observed on commercial poultry farms [44,61]. Additionally, studies from other countries have reported the dominance of serotypes other than the commonly identified *S*. Enteritidis and *S*. Typhimurium in humans, poultry, and poultry products [43,82–84].

In this study, a substantial proportion of poultry farms (77.3%) reported using antimicrobials for various purposes, including feed additives, treatment, and disease prevention. This aligns with findings from a related study that revealed approximately 84% of livestock farmers in Ethiopia use antibiotics on their farms. Tetracycline, aminoglycosides, and sulfonamide-trimethoprim were identified as the most frequently employed antibiotic classes [85]. The non-therapeutic use of antibiotics in animal agriculture, particularly in developing countries, is a growing concern. Antibiotics are often administered at sub-therapeutic doses as growth promoters to enhance livestock production and increase economic returns. However, this practice can inadvertently foster an environment that facilitates bacterial evolution and the emergence of antibiotic resistance [86].

The finding of relatively high resistance to tetracycline is consistent with studies conducted in India [87] and Turkey [88], which reported resistance rates of 100% and 93.34%, respectively. Sharma et al. [84] also demonstrated 100% resistance to tetracycline and 95.71% resistance to ampicillin in *Salmonella* isolates from retail chicken meat shops. Our results concur with previous findings that resistance to ampicillin, streptomycin, and tetracyclines was documented in *S*. Typhimurium strains isolated from diarrheic patients and food handlers in Kenya and northwestern Ethiopia [89,90]. The high resistance rate to ampicillin and tetracycline is likely the prolonged and extensive use of these antimicrobials in veterinary medicine and the poultry industry. This continuous use exerts selective pressure, contributing significantly to the development and spread of antibiotic resistance. Conversely, all *Salmonella* isolates in this study demonstrated 100% sensitivity to gentamicin and ciprofloxacin, consistent with findings from Adama and Modjo, Ethiopia [16], and Erzurum, Turkey [88]. However, these findings contrast with a previous report that indicated extensive drug resistance to ciprofloxacin in poultry and human samples from southern Ethiopia [26], Egypt [91], and Ghana [92]. The absence of resistance to ciprofloxacin and gentamycin in this study suggests that these medications remain effective treatment options for *Salmonella* in poultry production within the study area. The relatively low resistance to ciprofloxacin may be attributed to its infrequent use, high cost, and limited availability. Furthermore, the low resistance to gentamicin observed in *Salmonella* could be explained by the limited presence of inherent resistance mechanisms against aminoglycoside antibiotics.

In low-income countries, including Ethiopia, antimicrobials are often easily accessible without a prescription, resulting in indiscriminate use in poultry farming and human healthcare [93]. This widespread use of antimicrobials in the poultry industry has likely exerted continuous selection pressure on *Salmonella*, contributing to the emergence of antimicrobial-resistant strains. In addition, *Salmonella* possesses intrinsic defense mechanisms that enable its survival and the development of antibiotic resistance. These mechanisms include the enzymatic inactivation of antibiotics, the expulsion of drugs from the cell via an efflux pump, the modification of drug targets, and the reduction of membrane permeability. Moreover, extensive dissemination of mobile antimicrobial resistance genes through horizontal gene transfer can occur within and beyond poultry farm environments. This allows bacteria to persist even in the absence of direct selective pressure [94,95]. In this study, all *S*. Typhimurium and *S*. Enteritidis isolates exhibited MDR patterns with MAR index ranging from 0.45 to 0.55 across different sample sources. MAR index values exceeding 0.2 typically indicate that the isolates originate from

high-risk, heavily contaminated environments where antibiotic use is frequent [96]. The occurrence of MDR *Salmonella* underscores the potential role of chickens and poultry farms as significant reservoirs for human pathogens and antibiotic-resistant genes.

This study was subject to limitations, including its cross-sectional nature, exclusion of additional sources such as air, poultry products, and by-products, and inability to detect virulence and antibiotic resistance genes. Despite these limitations, this study underscores the role of the poultry industry in antimicrobial resistance dynamics and disease epidemiology within the broader regional context of the One Health framework. Longitudinal studies that incorporate appropriate sample size calculations, with a particular focus on diverse sample sources and farm-level clustering of chickens, could enhance our understanding of patterns of *Salmonella* transmission and evolution within and between farms. Additionally, such studies would allow a more comprehensive identification of farm-specific risk factors for *Salmonella* prevalence and antimicrobial resistance across sample sources. Future research should integrate fundamental microbiology with high-throughput multi-omics approaches to uncover the key drivers of antimicrobial resistance, elucidate the mechanisms of horizontal gene transfer, evaluate environmental influences and reservoirs, and map evolutionary pathways. Understanding these determinants can facilitate the development of novel strategies that extend beyond traditional antimicrobial stewardship efforts. Moreover, continuous evaluations of phenotypic resistance and the ability of the *Salmonella* strains to horizontally transfer resistance genes through minimum inhibitory concentration (MIC) determination and conjugation assays are substantial for accurately characterizing the bona fide resistance patterns in these strains.

## Conclusions

In summary, this study underscores the emergence of highly multidrug-resistant nontyphoidal *Salmonella* on poultry farms in northwestern Ethiopia, posing a significant public health threat. All *Salmonella* isolates exhibited complete resistance to tetracyclines and penicillins, antibiotics frequently used in veterinary practice. To effectively combat antibiotic resistance in this region, further efforts should focus on establishing an integrated surveillance system, promoting the prudent use of critical antibiotics, and exploring the potential of drug combinations and natural antimicrobial agents. *S.* Enteritidis emerged as the predominant serotype in chickens, humans, and farm environments. Given the zoonotic nature of these serotypes, high-resolution microbial genomics is crucial for understanding their transmission dynamics and identifying the sources of infection between humans and animals. Moreover, the presence of other animal species and improper foot baths were identified as potential risk factors for the spread of nontyphoidal salmonellosis. Therefore, implementing robust biosecurity measures, effective flock management practices, and controlling the presence of other animal species are essential for mitigating the spread of nontyphoidal salmonellosis in this region.

## Supporting information

**S1 Fig. Representative raw images collected during the study period.**
(PDF)

**S2 File. Semi-structured questionnaires used to gather detailed information from 22 poultry farms.**
(PDF)

## Acknowledgments

The authors would like to express their sincere gratitude to the farm owners and the Bahir Dar Agricultural Development Office for their cooperation and for providing the essential information for this study. We also extend our appreciation to the Microbiology Laboratory Unit of the Amhara Public Health Institute for their valuable technical assistance during laboratory activities throughout the experiment.

## Author contributions

**Conceptualization:** Azeb Bayu Mengistu, Mequanint Addisu Belete, Habtamu Tassew, Haregua Yesigat Kassa, Beyenech Gebeyehu Alemu, Hailehizeb Cheru Tegegne, Demelash Areda.

**Data curation:** Azeb Bayu Mengistu, Haregua Yesigat Kassa, Hailehizeb Cheru Tegegne.

**Formal analysis:** Azeb Bayu Mengistu, Haregua Yesigat Kassa, Hailehizeb Cheru Tegegne.

**Funding acquisition:** Habtamu Tassew.

**Investigation:** Azeb Bayu Mengistu, Mequanint Addisu Belete.

**Methodology:** Azeb Bayu Mengistu, Mequanint Addisu Belete, Habtamu Tassew, Haregua Yesigat Kassa, Beyenech Gebeyehu Alemu, Hailehizeb Cheru Tegegne, Demelash Areda.

**Software:** Azeb Bayu Mengistu.

**Supervision:** Mequanint Addisu Belete, Habtamu Tassew, Beyenech Gebeyehu Alemu, Demelash Areda.

**Validation:** Habtamu Tassew, Beyenech Gebeyehu Alemu, Demelash Areda.

**Writing – original draft:** Azeb Bayu Mengistu, Mequanint Addisu Belete, Habtamu Tassew.

**Writing – review & editing:** Azeb Bayu Mengistu, Mequanint Addisu Belete, Habtamu Tassew, Haregua Yesigat Kassa, Beyenech Gebeyehu Alemu, Hailehizeb Cheru Tegegne, Demelash Areda.

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
