## [Decision Letter · Decision Letter 0]

25 Jun 2025

Dear Dr. Belete,

Thank you for submitting your manuscript to PLOS ONE. After careful consideration, we feel that it has merit but does not fully meet PLOS ONE’s publication criteria as it currently stands. Therefore, we invite you to submit a revised version of the manuscript that addresses the points raised during the review process.

We look forward to receiving your revised manuscript.

Kind regards,

Csaba Varga, DVM MSc PhD

Academic Editor

PLOS ONE

Journal Requirements:

2. Please ensure that you refer to Figure 4 in your text as, if accepted, production will need this reference to link the reader to the figure.

3. Please upload a copy of Figure 3, to which you refer in your text on page 14. If the figure is no longer to be included as part of the submission please remove all reference to it within the text.

4. We note that Figure S1 includes an image of a participant / in the study.

Reviewers' comments:

Reviewer's Responses to Questions

**Comments to the Author**

1. Is the manuscript technically sound, and do the data support the conclusions?

Reviewer #1: Yes

Reviewer #2: Yes

Reviewer #3: Partly

2. Has the statistical analysis been performed appropriately and rigorously?

Reviewer #1: Yes

Reviewer #2: N/A

Reviewer #3: No

3. Have the authors made all data underlying the findings in their manuscript fully available?

Reviewer #1: Yes

Reviewer #2: Yes

Reviewer #3: No

4. Is the manuscript presented in an intelligible fashion and written in standard English?

Reviewer #1: Yes

Reviewer #2: Yes

Reviewer #3: Yes

Reviewer #1: 1. All scientific names should be italicised

2. The first sentence of the abstract must be revised based on the title

3. Revise your introduction; the second sentence is unnecessary

4. Study conducted only in one city (Bahir Dar) may not represent northwest Ethiopia, so revise your title and other parts based on the area where the study was conducted

Reviewer #2: This manuscript discuss the crossectional study of AMR in non-typhoidal Salmonella. Study has been informed and written well. Som eextreem words such as Salmonella is a devastating disease shoul dbe corrected. Association was observed through chisquare tests. Corrected chisquare test should be calculated for those celss with <5. For bivariate categories, it is better presented the Fisher's exact test rather than chi-square.

Reviewer #3: This manuscript reports on an important topic, however, my considered opinion is that this study was low powered and this affected the authors choice of statistical test. This study would have benefited from more rigorous statistical tests to assess the magnitude (strengths) and direction of association between the explanatory variables and the outcome variable. Also, in the design of the study, the authors should have considered the fact that Salmonella is intermittently shed by poultry and should have tested a large number of birds per farm rather than just 126 from 22 farms that had relatively large flock sizes.

Abstract

Line 43: Consider italicizing “Salmonella” before species and should be consistently done across the manuscript.

Line 52-54: Is this a conclusion from your study? The conclusions section of your manuscript needs to be improved. In my view, there are no conclusions drawn from the findings. Real conclusions should be provided here.

Line 58: “Occurrence” may not be a useful key word. Why include occurrence as a key word?

Introduction

Line 61: Be consistent with italicizing the word Salmonella throughout your manuscript.

Lines 63-65: I believe you are referring to human deaths. It would be nice to be specific.

Line 69: I believe it is not correct to start a statement with an acronym. You might consider writing NTS in full here.

Line 83-86: “associated illnesses originated from contaminated poultry” is different from contaminated poultry meat and eggs. Verify your facts and report this correctly. Is the statement on line 83-86 correct?

Line 112: S. Kentucky not Kentuck.

Line 118: epidemiologic not epidemiological.

Methods

Line 147: …July 2021)

Line 147-148: Consider rephrasing, though grammatically correct, please reconsider the use of the word subjects to refer to chickens in the study.

Line 149-150: How did you determine your sample size? I can see that your sample size could have affected your choice of statistical test.

Line 150: What was the minimum flock size in the small farms?

Line 153-157: Can you provide the questionnaire used as a supplementary file?

Line 169-171: It is important to state that the birds were sampled just once. This is important because salmonella shedding can be intermittent, and a negative test is not conclusive.

Line 175-177: What was the time between sample collection and submission to the lab? Was this consistent for all samples? How long were the samples stored in iceboxes?

Line 195: provide a reference for the guidelines.

Line 231: Outlined in what? Mention it before the citation.

Line 251-252: Why the Chi-square test? Couldn’t you perform analysis with logistic regression? Did your sample size affect your choice of statistical test? What were you explanatory variables (risk factors talked about)? These need to be described alongside your outcome variable. The Chi-square test does not give the magnitude (strength) of the association and the direction of the association. Explain to the reader the reasons for your choice of statistical test.

Results

Line 279: How strong was the association? What was the magnitude of the association? Provide the magnitude of the association. You cannot claim the association is strong without providing a measure of the strength of the association i.e. the odds ratio.

Line 284: Why use the word salmonella occurrence and not prevalence for consistency?

Line 292: Be consistent with your reporting of P-Values.

Line 308: Does sensitivity mean low resistance??

**Do you want your identity to be public for this peer review?** For information about this choice, including consent withdrawal, please see our Privacy Policy

Reviewer #1: No

Reviewer #2: No

Reviewer #3: No

---

## [Author Response · Author response to Decision Letter 1]

19 Aug 2025

Detailed Response to Editor and Reviewers’ Comments

We would like to express our sincere gratitude to the editors and reviewers for their valuable comments and constructive suggestions, which have greatly improved the manuscript. We have carefully considered all feedback and revised the manuscript accordingly. Below are our point-by-point responses to both major and minor comments. All changes made in the revised manuscript are highlighted with line numbers.

Journal Requirements

1. Please ensure that your manuscript meets PLOS ONE's style requirements, including those for file naming. The PLOS ONE style templates can be found at https://journals.plos.org/plosone/s/file?id=wjVg/PLOSOne_formatting_sample_main_body.pdf andhttps://journals.plos.org/plosone/s/file?id=ba62/PLOSOne_formatting_sample_title_authors_affiliations.pdf

Response: We ensured that our manuscript adheres to the style guidelines of PLOS ONE by referring to the provided links and following the submission requirements.

2. Please ensure that you refer to Figure 4 in your text as, if accepted, production will need this reference to link the reader to the figure.

Response: We sincerely apologize for the inconsistency in labelling the figures in the initial manuscript. The caption Figure 4 has been removed since the revised version contains three figures. Figure 3 is referenced appropriately in the text (lines 304, 309, 317, 325, 327).

3. Please upload a copy of Figure 3, to which you refer in your text on page 14. If the figure is no longer to be included as part of the submission please remove all reference to it within the text.

Response: As we mentioned in requirement 2, this problem arises from an issue with labelling the figures in sequence. We resolved the issue by renaming the figure to reflect its correct order.

4. We note that Figure S1 includes an image of a participant / in the study.

Response: Thank you. The authors have finally decided to eliminate the images of the participants due to the difficulty in obtaining consent from the individual within a short period.

Responses to comments from Reviewer #1

Comment: All scientific names should be italicised.

Response: Thank you for your valuable comment and for recognizing the issue. We addressed the comments in the revised manuscript.

Comment: The first sentence of the abstract must be revised based on the title.

Response: We have accordingly revised the first sentence of the abstract to ensure consistency (lines 28-31).

Comment: Revise your introduction; the second sentence is unnecessary.

Response: Thank you. The authors agree to remove the second sentence from the introduction as suggested.

Comment: The study was conducted only in one city (Bahir Dar) may not represent northwest Ethiopia, so revise your title and other parts based on the area where the study was conducted.

Response: We appreciate the reviewer’s suggestion to modify the title. We want to express our preference for using the phrase “Northwestern Ethiopia” in the title of the manuscript, underscoring the scarcity of similar research in this specific area, in contrast to the more extensively studied central and southern parts of the country. Additionally, this phrasing reflects the broader impact of Salmonella in comparable settings, considering the similarities in farm characteristics and production systems across the region. In response to the reviewer's suggestion, we have revised the title to “Multidrug-resistant non-typhoidal Salmonella enterica from chickens, farmworkers, and environments: One Health Implications from Northwestern Ethiopia” and have made necessary corrections by consistently mentioning “Bahir Dar city” throughout the manuscript (lines 33, 116, and 130).

Responses to comments from Reviewer #2

Comment: This manuscript discusses the cross-sectional study of AMR in non-typhoidal Salmonella. The study has been informed and written well. Some extreme words, such as Salmonella is a devastating disease, should be corrected. Association was observed through chi-square tests. The corrected chi-square test should be calculated for those cells with <5. For bivariate categories, it is better to present Fisher's exact test rather than chi-square.

Response: We appreciate the reviewer's perspective and their endorsement of the relevance of this study. In response to the first comment regarding the sentence ‘Salmonella is a devastating disease’, we have made a thorough correction in the second paragraph of the introduction (lines 66-67) to improve the scientific accuracy. Furthermore, the second issue related to statistical analysis has been addressed by implementing appropriate Fisher's exact analysis, following a careful evaluation of the data characteristics and in consideration of the reviewer’s valuable suggestions (lines 37, 250-252, and Table 2).

Responses to comments from Reviewer #3

Comment: This manuscript reports on an important topic, however, my considered opinion is that this study was low powered and this affected the authors choice of statistical test. This study would have benefited from more rigorous statistical tests to assess the magnitude (strengths) and direction of association between the explanatory variables and the outcome variable. Also, in the design of the study, the authors should have considered the fact that Salmonella is intermittently shed by poultry and should have tested a large number of birds per farm rather than just 126 from 22 farms that had relatively large flock sizes.

Response: The authors are grateful to the reviewer for recognizing the importance of this work and for their constructive feedback. We fully share the reviewer’s perspective and appreciate the opportunity to address these important concerns.

While we acknowledge that the number of Salmonella isolates obtained in this study was relatively low, we believe this does not undermine the study's validity, provided that the methodology and experimental procedures strictly follow established guidelines. Indeed, several peer-reviewed studies conducted in Ethiopia and elsewhere have similarly reported low isolation rates, which can be attributed to the intermittent shedding nature of Salmonella, environmental influences, and farm management practices.

DOI: 10.4315/0362-028x-72.10.2071 DOI: 10.1111/jfs.12585

https://doi.org/10.3382/ps/pew342 DOI: 10.1002/vms3.647

DOI: 10.4314/evj.v21i2.7 https://doi.org/10.1155/2022/8625636

https://doi.org/10.1155/2020/6425946

Our study included a total of 369 samples from various sources (animals, humans, and the environment), providing a broader One Health perspective. However, resource limitations, financial constraints, and logistical challenges collectively limited the number of cloacal samples collected per farm.

Regarding the statistical analysis, we carefully assessed the structure and distribution of the data. Due to the small sample sizes and the presence of expected cell frequencies below 5 in several contingency tables, Fisher’s exact test was chosen as the most appropriate and statistically sound method under the circumstances.

Given that this is the first report of its kind at a small-scale level in the study area, we acknowledge these limitations in the discussion section and recommend that future large-scale studies incorporate sample size calculations, parametric sampling, and regression-based analytical approaches to further explore the associations observed in this study (lines 516-521).

Comment: Line 43: Consider italicizing “Salmonella” before species, and it should be consistently done across the manuscript.

Response: Thank you. We consistently italicized the genus name "Salmonella" throughout the manuscript.

Comment: Line 52-54: Is this a conclusion from your study? The conclusions section of your manuscript needs to be improved. In my view, there are no conclusions drawn from the findings. Real conclusions should be provided here.

Response: The authors acknowledge the comment. The conclusion of the abstract has been revised intensely following the comments (lines 47-52).

Comment: Line 58: “Occurrence” may not be a useful key word. Why include occurrence as a key word?

Response: The Keyword has been corrected accordingly in the manuscript (line 53).

Comment: Line 61: Be consistent with italicizing the word Salmonella throughout your manuscript.

Response: Thank you. We consistently italicized the genus name "Salmonella" throughout the manuscript.

Comment: Lines 63-65: I believe you are referring to human deaths. It would be nice to be specific.

Response: We appreciate the reviewer's comments. This has been corrected in the revised manuscript by mentioning the immense role of Salmonella in causing diarrheal diseases and human deaths (line 58).

Comment: Line 69: I believe it is not correct to start a statement with an acronym. You might consider writing NTS in full here.

Response: Thank you. We define the acronym in the revised manuscript accordingly (line 63).

Comment: Line 83-86: “associated illnesses originated from contaminated poultry” is different from contaminated poultry meat and eggs. Verify your facts and report this correctly. Is the statement on line 83-86 correct? (lines 75-78).

Response: We appreciate the feedback. The comments are fully considered and corrected accordingly (lines 76-79).

Comment: Line 112: S. Kentucky not Kentuck.

Response: Revised accordingly (line 106).

Comment: Line 118: epidemiologic not epidemiological.

Response: We have made the corrections based on the feedback provided (line 112).

Comment: Line 147:…July 2021)

Response: Thank you. The modification has been done accordingly (line 140).

Comment: Line 147-148: Consider rephrasing, though grammatically correct, please reconsider the use of the word subjects to refer to chickens in the study.

Response: Thank you. The authors addressed the issue of the term "subject" with appropriate language and modifications to the sentences (lines 141-143).

Comment: Line 149-150: How did you determine your sample size? I can see that your sample size could have affected your choice of statistical test.

Response: The study includes multiple sample types, such as cloacal swabs from chickens, environmental samples (water, feed, boot swabs), and pooled stool samples from farm attendants.

We initially tried to determine the sample size using the appropriate single population (Thrusfield) formula, which is widely accepted for prevalence and risk factor studies. the authors, however, could not use the proposed sample size calculation because of the heterogeneous nature of the sample sources and the clustering of chickens within farms. The formula assumes a single, homogeneous population and does not consider the fundamental differences or interdependence among these sample types. As a result, we decided to determine the number of samples based on our available resources and laboratory capacity; in fact, this influenced our choice of statistical test. We acknowledged this limitation of the study by recommending longitudinal studies that include proper sample size calculations, especially focusing on diverse sample sources and farm-level clustering of chickens in the discussion and conclusion sections of the manuscript (lines 516-521).

Comment: Line 150: What was the minimum flock size in the small farms?

Response: We appreciate the comment. There were small farms that raised a minimum of 125 chickens during the sampling period. To ensure adequate sample representation from 22 poultry farms, we determined a target of 126 cloacal samples from live birds based on resource availability. To achieve this, we planned to collect a maximum of 6 live birds per farm, resulting in a proportional allocation of sampling effort across the included farms.

Comment: Line 153-157: Can you provide the questionnaire used as a supplementary file?

Response: Surely. The questionnaire is included as a supplementary file in the amended manuscript.

Comment: Line 169-171: It is important to state that the birds were sampled just once. This is important because Salmonella shedding can be intermittent, and a negative test is not conclusive.

Response: Thank you. We incorporated the information in the revised manuscript (lines 165-166).

Comment: Line 175-177: What was the time between sample collection and submission to the lab? Was this consistent for all samples? How long were the samples stored in iceboxes?

Response: All the previously mentioned questions are addressed by indicating the time interval between collection and submission, and how samples are stored or processed in the lab within the revised manuscript (lines 170-172).

Comment: Line 195: Provide a reference for the guidelines.

Response: Done accordingly (line 191).

Comment: Line 231: Outlined in what? Mention it before the citation.

Response: The criteria used for categorizing the level of resistance are mentioned before the citation in the revised manuscript (line 227).

Comment: Line 251-252: Why the Chi-square test? Couldn’t you perform analysis with logistic regression? Did your sample size affect your choice of statistical test? What were your explanatory variables (risk factors talked about)? These need to be described alongside your outcome variable. The Chi-square test does not give the magnitude (strength) of the association or the direction of the association. Explain to the reader the reasons for your choice of statistical test.

Response: Thank you for your valuable comment. We considered logistic regression as a more robust analytical approach to assess the degree of association between risk factors and the outcome variable. However, our sample size, lack of sample size determination formula, and expected cell frequencies below 5, particularly within subgroups, were insufficient to support meaningful multivariable logistic regression analysis. We have clarified the rationale for our statistical approach in the revised Data Analysis section (251-252).

Comment: Line 279: How strong was the association? What was the magnitude of the association? Provide the magnitude of the association. You cannot claim the association is strong without providing a measure of the strength of the association i.e. the odds ratio.

Response: Thank you. Apologize for the inconsistency in using technical terms irrelevant to the statistical test used in the study. We resolved the issue by omitting the unnecessary typographical errors.

Comment: Line 284: Why use the word Salmonella occurrence and not prevalence for consistency?

Response: We addressed the comment by consistently using the word "Prevalence" throughout the revised manuscript.

Comment: Line 292: Be consistent with your reporting of P-values.

Response: Thank you. The revision has been done accordingly.

Comment: Line 308: Does sensitivity mean low resistance??

Response: Thank you for your observation. We agree that the terminology requires clarification to avoid misinterpretation. In the manuscript, the term “antibiotic sensitivity” was used to indicate that the bacterial isolates were susceptible to the tested antibiotics. The phrase “low resistance” has been avoided to resolve such confusion.

---

## [Decision Letter · Decision Letter 1]

16 Sep 2025

Multidrug-resistant non-typhoidal Salmonella enterica from chickens, farmworkers, and environments: One Health Implications from Northwestern Ethiopia

PONE-D-25-25886R1

Dear Dr. Mequanint Addisu Belete,

We’re pleased to inform you that your manuscript has been judged scientifically suitable for publication and will be formally accepted for publication once it meets all outstanding technical requirements.

Kind regards,

Csaba Varga, DVM MSc PhD

Academic Editor

PLOS ONE

Additional Editor Comments (optional):

Reviewer #1:

Reviewer #2:

Reviewers' comments:

Reviewer's Responses to Questions

**Comments to the Author**

Reviewer #1: All comments have been addressed

Reviewer #2: (No Response)

2. Is the manuscript technically sound, and do the data support the conclusions?

Reviewer #1: Yes

Reviewer #2: Yes

3. Has the statistical analysis been performed appropriately and rigorously?

Reviewer #1: Yes

Reviewer #2: Yes

4. Have the authors made all data underlying the findings in their manuscript fully available?

Reviewer #1: Yes

Reviewer #2: Yes

5. Is the manuscript presented in an intelligible fashion and written in standard English?

Reviewer #1: Yes

Reviewer #2: Yes

Reviewer #1: 1. please check the references again, e.g. line 104 check the reference, cancel Taddesse,

2. I have other issues to raise

Reviewer #2: (No Response)

**Do you want your identity to be public for this peer review?** For information about this choice, including consent withdrawal, please see our Privacy Policy

Reviewer #1: No

Reviewer #2: No

---

## [Editor Report · Acceptance letter]

PONE-D-25-25886R1

PLOS ONE

Dear Dr. Belete,

I'm pleased to inform you that your manuscript has been deemed suitable for publication in PLOS ONE. Congratulations! Your manuscript is now being handed over to our production team.

Kind regards,

on behalf of

Dr. Csaba Varga

Academic Editor

PLOS ONE